# Adhesion G Protein-Coupled Receptor *Gpr126* (*Adgrg6*) Expression Profiling in Diseased Mouse, Rat, and Human Kidneys

**DOI:** 10.3390/cells13100874

**Published:** 2024-05-18

**Authors:** Peter Kösters, Salvador Cazorla-Vázquez, René Krüger, Christoph Daniel, Eva Vonbrunn, Kerstin Amann, Felix B. Engel

**Affiliations:** 1Department of Nephropathology, Experimental Renal and Cardiovascular Research, Institute of Pathology, Friedrich-Alexander-Universität Erlangen-Nürnberg (FAU), 91054 Erlangen, Germany; peter.koesters@fau.de (P.K.); s_cazorla@hotmail.com (S.C.-V.); christoph.daniel@uk-erlangen.de (C.D.); eva.vonbrunn@uk-erlangen.de (E.V.); kerstin.amann@uk-erlangen.de (K.A.); 2Department of Nephrology and Hypertension, Universitätsklinikum Erlangen, Friedrich-Alexander-Universität Erlangen-Nürnberg (FAU), 91054 Erlangen, Germany; rene.krueger@uk-erlangen.de

**Keywords:** adhesion GPCR, kidney disease, *Gpr126*, parietal epithelial cell, collecting duct, urothelium, RNAscope, human, focal segmental glomerulosclerosis

## Abstract

Uncovering the function of understudied G protein-coupled receptors (GPCRs) provides a wealth of untapped therapeutic potential. The poorly understood adhesion GPCR *Gpr126* (*Adgrg6*) is widely expressed in developing kidneys. In adulthood, *Gpr126* expression is enriched in parietal epithelial cells (PECs) and epithelial cells of the collecting duct and urothelium. Whether Gpr126 plays a role in kidney disease remains unclear. Here, we characterized *Gpr126* expression in diseased kidneys in mice, rats, and humans. RT-PCR data show that *Gpr126* expression is altered in kidney disease. A quantitative RNAscope^®^ analysis utilizing cell type-specific markers revealed that *Gpr126* expression upon tubular damage is mainly increased in cell types expressing *Gpr126* under healthy conditions as well as in cells of the distal and proximal tubules. Upon glomerular damage, an increase was mainly detected in PECs. Notably, *Gpr126* expression was upregulated in an ischemia/reperfusion model within hours, while upregulation in a glomerular damage model was only detected after weeks. An analysis of kidney microarray data from patients with lupus nephritis, IgA nephropathy, focal segmental glomerulosclerosis (FSGS), hypertension, and diabetes as well as single-cell RNA-seq data from kidneys of patients with acute kidney injury and chronic kidney disease indicates that *GPR126* expression is also altered in human kidney disease. In patients with FSGS, an RNAscope^®^ analysis showed that *GPR126* mRNA is upregulated in PECs belonging to FSGS lesions and proximal tubules. Collectively, we provide detailed insights into *Gpr126* expression in kidney disease, indicating that GPR126 is a potential therapeutic target.

## 1. Introduction

Together, acute kidney injury (AKI) and chronic kidney disease (CKD) represent the fastest growing pathology worldwide. The prevalence of CKD in many countries is >10% [1]. As there are currently no effective therapies to restore kidney function, it is important to establish diagnostic markers/therapies that prevent, reduce, or even reverse kidney damage. Adhesion G protein-coupled receptors (aGPCRs) represent the second largest class of the GPCR superfamily, but they are poorly understood. aGPCRs are characterized by an extended N-terminal fragment (NTF) containing a multitude of adhesion-like domains, which is linked via the so-called GAIN domain to the C-terminal fragment (CTF) containing a 7-transmembrane domain (7TM). The GAIN domain contains a GPCR proteolysis site (GPS) motif through which most of the aGPCRs are cleaved into NTF and CTF [2]. Cleavage results in the release of a tethered agonist for CTF activation, so-called *Stachel* sequences [3,4], and of the NTF, which can act in a non-cell autonomous fashion [3,5].

GPCRs are involved in a large number of human diseases and are currently targeted by ~34% of all approved drugs by the US Food and Drug Administration [6]. Therefore, uncovering the function of understudied GPCRs, such as aGPCRs, provides a wealth of untapped therapeutic potential. Notably, a variety of kidney diseases are associated with altered aGPCR expression including CKD, diabetic nephropathy, and lupus nephritis [7]. A direct link between aGPCRs and kidney development has been shown for Celsr1 (ureteric bud (UB) branching) [8]. An analysis of mice with a point mutation in Celsr1 (Celsr1^Crsh/Crsh^) at embryonic day E13.5 showed that mutant mouse kidneys exhibit a smaller, less complex ureteric tree containing significantly reduced numbers of total ureteric bud tips, tree segments, and branch points. In addition, it has been shown that *Adgrg3* (*Gpr97*) plays a role in AKI [9]. Adgrg3 expression was upregulated on mRNA and affected the protein level in the mouse kidney post ischemia/reperfusion, mainly in PTs and DTs. Importantly, Adgrg3 expression was also upregulated in the kidneys of patients with biopsy-proven acute tubular necrosis, which presents with AKI and is one of the most common causes of AKI. Notably, in AKI models, Adgrg3-deficient mice had significantly less renal injury and inflammation and better function than wild-type mice. Based on a microarray-based analysis and a Western blot analysis, as well as a series of in vitro assays including the siRNA-mediated knockdown of Adgrg3 expression, the authors showed that Adgrg3 mediates its protective function via semaphorin 3A signaling and concluded that “Adgrg3 is an important mediator of AKI, and pharmacologic targeting of Gpr97-mediated Sema3A signaling at multiple levels may provide a novel approach for the treatment of AKI” [9]. In addition, polycystin-1, a major regulator of kidney development and disease, contains a GAIN domain, which is otherwise only found in aGPCRs [10].

Recently, we showed that the aGPCR *Gpr126* (*Adgrg6*) is widely expressed in developing kidneys in zebrafish, mice, and humans [11,12]. In the adult kidney, *Gpr126* expression is enriched in parietal epithelial cells (PECs) and epithelial cells of the collecting duct (CD) and urothelium [11]. Knockout studies revealed that Gpr126 plays an important role in development and/or repair in a variety of tissues/organs, such as the heart [5], sciatic nerve [13,14,15], and cartilage [16]. In addition, genomic and transcriptomic patient-based studies such as whole-exome sequencing and genome-wide association studies revealed that *GPR126* mutations are associated with human diseases like adolescent idiopathic scoliosis and lung disease [2]. Furthermore, it has been shown that Gpr126 function can be modulated in vivo by the peptide FT(23-50), which induces Gpr126-dependent signaling in mice [17], and miR27a/b, which can be utilized to reduce *Gpr126* expression [18]. Yet, no data are available regarding the role of Gpr126 in kidney disease. Therefore, we aimed to determine the cellular expression pattern of *Gpr126* in kidney disease. These experiments will provide valuable information to determine whether Gpr126 is a potential therapeutic target in kidney disease and whether the effects that the inducible, kidney-specific deletion of Gpr126 have on kidney function should be investigated in kidney injury models.

## 2. Materials and Methods

### 2.1. Tissues

Studies on *Gpr126* expression were performed on formalin-fixed paraffin-embedded (FFPE) tissues of different species (mouse, rat, and human). All samples were provided by the Department of Nephropathology (Friedrich-Alexander-Universität Erlangen-Nürnberg, Germany). For in situ hybridization studies, sections of 4 µm thickness were cut.

### 2.2. Human Material

The data obtained in this study were processed according to the principles of scientific practice, and all patient samples were analyzed anonymously. Kidney biopsies from patients with primary focal segmental glomerulosclerosis (FSGS) were analyzed. The use of FFPE human tissue material was approved by the Ethics Committee of the Friedrich-Alexander-Universität Erlangen-Nürnberg for the archive of the Department of Nephropathology (Re. No.4415 and Re. No.22-150-D), waiving the need for informed consent due to its retrospective nature.

### 2.3. Animal Housing and Experimentations

Mice and rats used for animal experiments were maintained in a specific pathogen-free facility in a temperature- and light-controlled environment and had ad libitum access to chow and water. Animals were kept on classical aspen wood bedding (sniff Spezialdiäten GmbH, Soest, Germany) and housed in macrolon cages (Tecniplast Deutschland GmbH, Hohenpeißenberg, Germany). The experimental protocol for the animal studies was approved by the German regional committee for animal care and use, which is equivalent to the US IACUC, and was authorized by the governmental department (“Regierung von Unterfranken” angiotensin II model and unilateral ureteral obstruction model (UUO) permit number: 54-2532.1-53/12; podocyte injury model permit number: 55.2-2532-2-225; ischemia/reperfusion model (I/R) permit numbers: 54-2532.1-30/11 and 55.2.2-2532-2-1127). The animal studies were performed in strict accordance with the German Welfare Act and adhered to the ARRIVE and BJP guidelines.

#### 2.3.1. Mouse Unilateral Ureteral Obstruction Model (UUO)

Fibrotic remodeling in the ligated kidney induced by urinary obstruction serves as a model for interstitial fibrosis occurring in chronic renal failure. Prior to the induction of the UUO model in 12–15-week-old male C57BL/6J mice, 0.1 mg/kg of body weight buprenorphine (Buprenovet, Elanco Deutschland GmbH, Bad Homburg, Germany) was subcutaneously administered as an analgesic before opening the abdominal cavity, and then the abdominal cavity was opened under isoflurane anesthesia, and the right ureter was ligated with a Ethicon coated Vicryl 5-0 (Johnson & Johnson International, New Brunswick, NJ, USA) suture directly at the renal pelvis. The wound was sutured in two layers (Ethicon-coated Vicryl 6-0, Johnson &Johnson International). Analgesia was maintained for 3 days after surgery (0.05 mg buprenorphine/kg body weight). The contralateral untreated kidney was used as a control and was harvested together with the ureter-ligated one at the end of experiment after 7 days, again with a combination of analgesia with 0.1 mg/kg of buprenorphine and isoflurane anesthesia.

#### 2.3.2. Mouse Angiotensin II Model (AngII)

Angiotensin II (Ang II; Sigma Aldrich Chemie GmbH, Steinheim, Germany) infusion causes systemic vasoconstriction and, in the kidney, a particular reduction in blood flow to the juxtamedullary and medullary areas of the kidney. As a result, tubulointerstitial damage occurs in these areas. This hypertension model was induced by the subcutaneous implantation of osmotic minipumps (Alzet 2004 (Alzet osmotic minipumps, Cupertino, CA, USA), filled with 3 µg kg^−1^ min^−1^ of Ang II diluted in PBS (Sigma Aldrich Chemie GmbH)) in 12–14-week-old C57bl/6J males. The mice were anesthetized with isoflurane, analgesia was subcutaneously administered with buprenorphine (0.05 mg/kg body weight), the fur was dorsally shaved on the right flank, the skin was disinfected, a skin incision of approximately 1 cm was made, and a subcutaneous pocket was prepared using blunt scissors. The filled osmotic minipump was then placed in the prepared skin pocket, and the skin was closed with single-button sutures (Ethicon-coated Vicryl 6-0, Johnson &Johnson International). On the following day, analgesia was again subcutaneously administered with buprenorphine (0.05 mg/kg body weight). The pumps were left in place for a period of 4 weeks. At the end of the experiment, the mice were sacrificed, and the kidneys were harvested for analysis.

#### 2.3.3. Rat Podocyte Injury Model (hDTR)

In contrast to humans, wild-type rats do not have a receptor for diphtheria toxin. In this model of podocyte injury, homozygous F344/NHsd-*Tg(NPHS2-HBEGF)* transgenic rats of both sexes expressing the human diphtheria toxin (DTx) receptor under the podocin promoter (kindly provided by Prof. R. Wiggins, University of Michigan; Wharram et al., 2005; Sato et al., 2009) [19,20] were administered 12.5 ng of DTx (Merck KGaA, Darmstadt, Germany)/kg of body weight that was intravenously diluted in PBS supplemented with 0.1 mg/mL of rat albumin (Sigma Aldrich Chemie GmbH) as a carrier under isoflurane anesthesia. Survival biopsy was performed 21 days after toxin administration after analgesia with 0.05 mg of buprenorphine/kg of body weight and isoflurane anesthesia by removing the upper pole of the left kidney with a scalpel and immediately closing the wound with a collagen sponge (KOLLAGENresorb, Resorba Medical GmbH, Nürnberg, Germany). Finally, the experiment was terminated either on day 21 or day 42 after model induction, and the kidneys were harvested for analysis.

#### 2.3.4. Rat Ischemia/Reperfusion Model (I/R)

For the induction of I/R, 12-week-old male dark agouti rats were first unilaterally nephrectomized. This kidney was used as a control. After isoflurane anesthesia and analgesia with 0.05 mg of buprenorphine/kg of body weight, a flank incision was made to induce the right I/R model after shaving and disinfection, the vessels of the right kidney were ligated, and the kidney was then removed. After closure of the abdominal wall and skin, the left flank was opened again 7 days later under isoflurane anesthesia and buprenorphine analgesia, and the renal artery and vein were carefully dissected and separated. The renal artery was disconnected from the blood supply for 30 min with an atraumatic clamp (Fine Science Tools GmbH, Heidelberg, Germany). During this process, animals were maintained at 37 °C body temperature on a heated operating table (Hugo Sachs Elektronic, March, Germany). After removal of the clamps, reperfusion of the kidney was checked for 2 min, and the abdominal wall and skin were closed with 2 sutures (Ethicon-coated Vicryl 6-0, Johnson &Johnson International). Animals were sacrificed after anesthesia and analgesia 3 days after model induction, and the kidneys were removed for further analysis. In a second experiment, we investigated how *Gpr126* expression changes over time after I/R induction. For this purpose, groups of 8 animals were sacrificed 10 min, 6 h, 24 h, 3 days, 5 days, and 8 weeks after reperfusion. Serum was collected at endpoints to determine serum urea as a surrogate marker of kidney function.

### 2.4. RT-PCR and Nanostring Multiplex mRNA Analysis

For real-time PCR (RT-PCR), individual kidneys were harvested from different diseased models (mouse UUO, mouse Ang-II, and rat I/R), grossly cut into pieces, and homogenized in 600 μL RLT Plus lysis buffer (1053393; Qiagen GmbH, Hilden, Germany) using an automatic TissueLyser II (85300; Qiagen GmbH) at 30 Hz for 4 min. RNA extraction was then performed with an RNeasy Mini Kit (74134; Qiagen GmbH), and cDNA was generated from sample volumes containing 1 μg RNA using M-MLV reverse transcriptase (M1302; Sigma-Aldrich, St. Louis, MI, USA) protocol according to the manufacturer’s instructions. To analyze *Gpr126* gene expression levels relative to endogenous *Gapdh*, RT-PCR assays were performed in triplicate for each biological replicate using SYBR^®^ Green (172-5271; Bio-Rad Laboratories, Inc., Hercules, CA, USA) in a CFX Connect^®^ real-time thermocycler (1855201; Bio-Rad) and the following primers: mouse *Gapdh* (5′-TGTCAAGCTCATTTCCTGGTATG-3′ and 5-GGCCTCTCTTGCTCAGTGTC-3′), rat *Gapdh* (5′-CATGGCCTTCCGTGTTCCTA-3′ and 5′-ACTTGGCAGGTTTCTCCAGG-3′), mouse *Gpr126* (5′-CCAGCTGGGTATGTGTAGCG-3′ and 5′-AGCCAGGCTTGTTTGGACAT-3′), and rat *Gpr126* (5′-TTTCTCCTAAGAGAATTTCTTCAACAA-3′ and 5′-TGATTGATGTCTGCTCTCCAC-3′). Relative *Gpr126* gene expression was calculated based on the 2^−ΔΔCq^ method [21] using *Gapdh* as the housekeeping gene. In the second I/R model experiment, a *Gpr126* expression analysis was performed using multiplex mRNA expression analysis using a custom code set (NanoString Technologies, Seattle, WA, USA) as described previously [22].

### 2.5. Routine Histology

Routine hematoxylin–eosin (H&E) and periodic acid–Schiff (PAS) staining was carried out using tissue processing machines (HistoCore Spectra ST; Leica, Wetzlar, Germany). Imaging was performed using a brightfield microscope (BX60; Olympus K.K., Shinjuku, Tokio, Japan).

### 2.6. RNAscope^®^ In Situ Hybridization

For the localization of cells expressing *Gpr126*, ACD’s RNAscope^®^ Multiplex Fluorescent Reagent Kit v2 (323100; Advanced Cell Diagnostics, Hayward, CA, USA) was used following the manufacturer´s instructions. mRNA of *Gpr126* was detected using species-specific target probes designed by ACD (Mm-Adgrg6, 472251; Rn-Adgrg6, 587651; Hs-Adgrg6, 480212) and labeled with the fluorophore Opal^®^ 570 (FP1488001KT; Akoya Biosciences, Marlborough, MA, USA). After the hybridization steps, the sections were co-stained with the cellular markers Claudin-1 (1:100; ab15098; Abcam, Cambridge, UK), Megalin (1:200; HPA005980; Atlas Antibodies AB, Bromma, Sweden), E-cadherin (1:200; BD 610181; BD Transduction Laboratories, Lexington, KY, USA), and Aquaporin-2 (1:100; SAB5200110; Sigma-Aldrich, St. Louis, MO, USA) for the identification of kidney-specific cells and tubular segments. Right after C1 signal development, the slides were washed in 50 mM of Tris buffer (pH 7.4) two times, and 0.5% bovine serum albumin (BSA) (8076; Carl Roth, Karlsruhe, Germany) in Tris was put on as a blocking solution for one hour at room temperature. Primary antibody incubation was carried out overnight at 4 °C in the dark. After washing three times in Tris, the markers were detected using Alexa Fluor^®^ 488 donkey anti-rabbit (1:200; A21206; Life Technologies, Eugene, OR, USA) and Alexa Fluor^®^ 647 donkey anti-mouse (1:200; A31571; Life Technologies). Both primary and secondary antibodies were diluted in 0.5% BSA in Tris. After counterstaining for 30 s with DAPI (4′,6′-diamidino-2-phenylindole) solution provided in the kit, the sections were mounted with a combination of DAPI solution (40 μg/mL in H_2_O) with Mowiol mounting medium (40 g Mowiol, 475904, Calbiochem, La Jolla, CA, USA in 200 mL 0.2 M Tris-HCl, pH 8.5; 100 mL glycerin; 8.75 g DABCO, Sigma, D-2522 + 2 mL DAPI solution). Imaging was conducted within two weeks using a laser scanning confocal microscope (800 LSM; Carl Zeiss AG, Oberkochen, Germany), producing single-layer images.

### 2.7. Image Analysis

All fluorescence images were processed in ImageJ (version number 1.53c) with the Fiji (version number 2.1.0) extension package [23] and Adobe^®^ Photoshop CC 2018.

### 2.8. scRNA-Seq Analysis

The scRNA-seq data of healthy living donors, individuals with acute kidney injury (AKI), and individuals with chronic kidney disease (CKD) published by Lake BB et al. are available as Seurat-object count data and metadata. These data were obtained from the Kidney Tissue Atlas (https://atlas.kpmp.org/, accessed on 18 July 2023) and processed using Seurat (version 0.6.0). To identify the gene expression of *GPR126* across all cell types, UMAPs were constructed for each individual and condition via FeaturePlot from Seurat (version 0.6.0). Raw counts were used to compute a pseudo bulk object. This was then normalized to Counts per Million (CPM) using “NormalizeData()” with parameters “scale.factor = 1 × 10^6^” and “normalization.method = “RC”” for each cell type and individual. We analyzed the normalized pseudo bulk expression of RNA from *GPR126* in 13 different cell types (podocyte (POD), parietal epithelial cell (PEC), proximal tubule (PT), descending thin limb (DTL), ascending thin limb/thick ascending limb (ATL/TAL), thick ascending limb (TAL), distal convoluted tubule (DCT), connecting tubule (CNT), principal cell (PC), intercalated cell (IC), endothelial cell (EC), Interstitial, and Immune) and filtered the data to include only samples from the three conditions (living donor, AKI, and CKD) with at least ten cells and three samples per cluster and condition. Thus, we excluded POD and DTL cell types because only one LD sample was left after filtering. We plotted both the general expression of the three conditions (LD, AKI, and CKD) and the individual and cell type-specific expression as a data point on top of the respective boxplot using ggpubr (version 0.6.0).

### 2.9. Statistical Analysis

Statistical analysis of RNAscope^®^ and Nanostring data as well as correlation analysis using the Spearman test were performed using GraphPad^®^ Prism 5. To compare two groups, an unpaired two-sample *t*-test was used. For multiple comparisons, ANOVA and Dunn’s test were used. Differences were considered statistically significant when the *p*-value was ≤ 0.05. When showing mean values, deviations are displayed as standard deviation (SD).

For the analysis of NephroSeq data (accessed on 29 April 2024) and single cell RNA-seq data, a multi-comparison between the three groups was performed using a two-sided *t*-test or the “dunn_test” function in rstatix (version 0.7.0) (https://cran.r-project.org/web/packages/rstatix/index.html, accessed on 10 July 2023), respectively.

## 3. Results

### 3.1. Gpr126 Expression Is Increased in a Model of Kidney Fibrosis in a Cell Type-Specific Manner

Recently, we found that *Gpr126* is expressed during kidney development [11,12]. To determine whether *Gpr126* expression is altered during kidney disease, we performed RT-PCR analyses measuring the *Gpr126* RNA levels in kidneys from animal models with mild (Ang II treatment, model of hypertensive nephropathy, mouse [24]) and severe kidney damage (I/R, model of AKI, rat [25]; UUO, model of interstitial fibrosis occurring in chronic renal failure, mouse [26]). While the Ang II treatment resulted in a modest increase (1.5-fold), I/R and UUO resulted in a marked upregulation of *Gpr126* expression (2.5- and 8-fold, respectively, Appendix A). These data show that *Gpr126* expression is modulated in disease, whereby the *Gpr126* expression levels depend on the type of injury.

The model of UUO is widely used to elucidate the mechanisms involved in obstructive nephropathy as well as progressive renal fibrosis, which is an irreversible condition that can arise from virtually any kidney pathology [26,27]. Therefore, we utilized this model to determine whether increased *Gpr126* expression levels in injured kidneys are due to an increased expression in cells known to express *Gpr126* or due to *Gpr126* expression in other cell types. The left ureter of C57BL/6J mice was clamped, inducing a rise in hydrostatic pressure and a dilation of the tubular system [28]. Seven days later, both kidneys were removed. Given the lack of commercially available validated antibodies for Gpr126, *Gpr126* expression was determined via a microscopic analysis of FFPE tissue using the RNAscope^®^ in situ hybridization technique in combination with immunofluorescent staining utilizing renal cell type-specific antibodies, comparing *Gpr126* expression in the left (clamped) and right kidneys (control) (Figure 1a). Under healthy conditions, *Gpr126* is typically expressed in PECs, the CD, and the pelvic urothelium in the mouse kidney [11,29]. PECs can be detected by their typical morphology and location, lining the Bowman’s capsule. CDs were identified by the colocalization of the CD-specific markers Aqp2 and E-cadherin [30]. The renal urothelium was recognized by E-cadherin staining and its localization in the renal pelvis (Figure 1b) [31]. In the UUO-affected kidneys, we observed that *Gpr126* expression is mainly increased in the cell types that also express *Gpr126* under healthy conditions (PECs, CD, and urothelium). However, an RNAscope^®^ signal could also be detected sporadically in the cortex in nephron segments, namely the distal tubule (DT, E-cadherin-positive but in contrast to the Aqp2-negative CD) and the proximal tubule (PT), which do not express *Gpr126* under healthy circumstances (Figure 1b).

For a quantitative analysis, the number of RNAscope^®^ signal dots contained in each cell was manually counted. Using ImageJ’s *cell counter* function, the cells belonging to a certain cell type or tubular segment were binned into seven groups based on the number of RNAscope^®^ signal dots within each cell (group 1: 0 dots/cell; group 2: 1 dot/cell; group 3: 2 dots/cell; group 4: 3 dots/cell; group 5: 4 dots/cell; group 6: ≥5 dots/cell; group 7: cells containing a dot cluster). By analyzing at least three different areas in the cortex, outer stripe of the outer medulla (OSOM), inner stripe of the outer medulla (ISOM), inner medulla (IM), and pelvic urothelium, we could gather detailed insights into the expressional changes in tubular segments and cell types associated with the named kidney regions.

The percentage of cells containing at least three signal dots is shown in Figure 2a,b. Changes were the most significant in the PECs (6.5 ± 5.85% to 40.1 ± 14.39%; fold change 6.1; *p* < 0.01; n = 4), DT of the cortex (0.6 ± 1.06% to 13.5 ± 4.61%; fold change 22.5; *p* < 0.01; n = 3), cortical CD (10.5 ± 6.57% to 46.1 ± 5.62%; fold change 4.4; *p* < 0.01; n = 3), CD in the ISOM (6.5 ± 3.02% to 34.2 ± 8.11%; fold change 5.3; *p* < 0.01; n = 3), and the pelvic urothelium (27.3 ± 8.97% to 76.0 ± 4.48%; fold change 2.8; *p* < 0.01; n = 3). Notably, the cortical DTs stand out with the highest fold change of all cell types analyzed (22.5-fold increase). However, it has to be considered that this nephron segment only expressed very little *Gpr126* in the control kidney (0.6 ± 1.06%), possibly mitigating the relevance of this increase. Taken together, the RNAscope^®^ data confirm the *Gpr126* upregulation detected by RT-PCR in the UUO model, which occurs in distinct parts of the nephron (mainly PECs, CD, urothelium).

To validate the correct identification of PECs based on the morphological criteria in Figure 1b, on sections from the same samples, we conducted immunofluorescent co-staining on RNAscope^®^-treated tissue sections using the PEC marker Claudin-1 [32] (Figure 2c). The RNAscope^®^ signal was detected in Claudin-1-positive cells, and the analysis of the signal in the UUO-affected kidney compared to the contralateral kidney confirmed that *Gpr126* expression is increased in PECs upon injury. To assess *Gpr126* expression in distal nephron segments and PTs, anti-Megalin and anti-E-cadherin antibodies were utilized [33]. As expected, E-cadherin- and Megalin-positive staining were mutually exclusive. As observed in Figure 1b and Figure 2a, the upregulation of *Gpr126* expression in the UUO-affected kidney was more evident in the E-cadherin-positive distal nephron segments compared to the Megalin-positive PTs.

Collectively, these data indicate that *Gpr126* expression is mainly upregulated in UUO-affected kidneys in PECs, distal nephron segments (DT and CD), as well as the urothelium.

### 3.2. Gpr126 Expression Is Markedly Increased in PECs in a Model of Glomerular Damage

To determine whether *Gpr126* expression is also altered in other kidney disease models, the transgenic rat model of DTx-induced podocyte depletion was analyzed, which represents glomerular disease. In our study, F344-*Tg*(*NPHS2-HBEGF*) rats were intravenously injected with either 12.5 ng of DTx/kg of body weight (BW) or 0.0 ng of DTx/kg of BW (sham) and sacrificed after 21 or 42 days. Like in the murine UUO model, *Gpr126* was analyzed using RNAscope^®^ technology (Figure 3a). Notably, in the sham-treated animals, we found *Gpr126* to be expressed in a similar fashion as previously described for normal untreated adult mouse and human kidneys [11], with the highest expression levels being found in PECs, CD, and the urothelium (Figure 3b,c). However, in contrast to human and mouse, we could also find low levels of *Gpr126* expression in PTs (PT OSOM: 15.9 ± 1.71%; n = 3). Interestingly, *Gpr126* expression in the CD increased in the direction of the papilla (CD cortex: 2.7 ± 0.94%; n = 3) (CD IM: 28.2 ± 6.09%; n = 3). We conclude that the expression pattern of *Gpr126* is conserved in healthy adult human, mouse, and rat kidneys. Subsequently, we analyzed *Gpr126* expression in the diseased kidneys. Twenty-one days post injection (dpi) of DTx, no significant changes in *Gpr126* expression could be observed (19.6 ± 13.33% to 18.8 ± 7.60%; n = 4). However, after 42 days, there was a marked increase in *Gpr126* expression in the PECs of the DTx-treated rats (13.8 ± 4.05% vs. 60.9 ± 16.08%; fold change 4.4; *p* < 0.01; n = 4) (Figure 3b,d). In the remaining kidney structures that were analyzed (PT, DT, CD, and urothelium), no significant changes in Gpr126 expression were found (Figure 3c,e). We conclude that *Gpr126* upregulation in the rat model of podocyte depletion is limited to PECs and occurs between 21 dpi and 42 dpi. 

Analogous to the mouse UUO model (Figure 2c), on sections from the same samples used for the quantification in Figure 3c, we performed immunofluorescent co-staining on RNAscope^®^-treated tissue sections to confirm the correct identification of PECs and PT based on morphological criteria (Figure 4). Claudin-1 staining colocalizes with cells that were previously identified as PECs by their typical morphology and localization in the Bowman´s capsule. Notably, the PECs of the DTx-treated animals remained Claudin-1 positive. In addition, PTs could be identified by Megalin (Figure 4).

### 3.3. GPR126 Expression Is Increased in Patients with AKI and CKD

Our data show that *Gpr126* expression is upregulated in the UUO (mouse; Figure 1 and Figure 2) and hDTR (rat; Figure 3 and Figure 4) kidney disease models. The RT-PCR analyses further showed that *Gpr126* is also upregulated upon I/R (rat) and Ang II-treatment (mouse; Appendix A). 

To determine whether *GPR12*—*6* expression is also altered in human kidney disease, we analyzed the microarray data provided by the comprehensive online resource NephroSeq v4 (www.nephroseq.org, accessed on 29 April 2024) for five diseases: lupus nephritis (Lupus_Glom; control: n = 14; diseases: n = 32), IgA nephropathy (IgAN_TubInt; control: n = 6; diseases: n = 25 and IgAN_Glom; control: n = 6; diseases: n = 27), FSGS (FSGS_Glom; control: n = 16; diseases: n = 8), hypertension (Hypertension_Glom; control: n = 4; diseases: n = 14), and diabetes (Diabetes_TubInt; control: n = 10; diseases: n = 12) (Appendix A). Notably, the NephroSeq data are highly variable within the individual disease groups (e.g., in regard to the affected glomeruli, the extent of sclerosis/fibrosis, the glomerular filtration rate, age, and gender), and *GPR126* expression was not assessed in a cell type-specific manner but in glomeruli (Glom) or the tubulointerstitium (TubInt).

An analysis of the Lupus_Glom data revealed no significant change in *GPR126* expression in diseased glomeruli compared to the healthy controls (Appendix A). Lupus nephritis is caused by autoantibodies directed against nuclear and cellular antigens, leading to immune complex formation and the accumulation of immune complexes in glomeruli. Patients often lack overt signs of kidney disease. Kidney damage in lupus nephritis is very variable (class I to class VI) [34]. The Lupus_Glom data were mainly derived from patients classified into class II to class IV, with a glomerular filtration rate (GFR) of ~60 mL/min/1.73 m^2^ (14 to 104) compared to ~75 mL/min/1.73 m^2^ in healthy living donors (60 to 97) (Appendix A). Also, the analysis of the IgAN_TubInt and IgAN_Glom data showed no significant change in *GPR126* expression. IgA nephropathy is a kidney disease induced by immunoglobulin A (IgA) accumulation in the kidney, causing inflammation, similar to lupus nephritis [35]. Also, here, the GFR was not markedly affected in the patients. In contrast to lupus nephritis and IgA nephropathy, an analysis of the FSGS_Glom data showed a trend towards increased *GPR126* expression in the glomeruli of patients with FSGS. FSGS is characterized by podocyte loss followed by the induction of repair mechanisms and, subsequently, glomerular scarring [36]. Thus, these data fit the increased *Gpr126* expression in PECs observed in the rodent model of podocyte depletion (Figure 3). Notably, the *GPR126* expression levels correlate with the percentage of affected glomeruli (Appendix A). While we observed a modest increase in *Gpr126* expression in the mouse model of hypertensive nephropathy (1.5-fold, Appendix A), an analysis of the Hypertension_Glom data revealed a significant decrease in *GPR126* expression (Appendix A). Hypertension causes damage to all blood vessels in the body over time, including the kidney [37], which markedly affects the GFR (~42 mL/min/1.73 m^2^). However, data for only four controls were included (tumor nephrectomy), and the majority of the patients received nephroprotective medication. Finally, a significant marked increase in *GPR126* expression was detected in the tubulointerstitium of kidney biopsies taken from patients with diabetes with markedly reduced GFRs (~20 mL/min/1.73 m^2)^. These data confirm the data from the animal models that *GPR126* expression is differentially affected by different types of kidney disease.

In addition to the NephroSeq data, we utilized the recently published human scRNA-seq data from the study by Lake et al. [38]. An analysis of the expression pattern of *GPR126* in kidney samples of healthy living donors revealed that it was in agreement with our previous findings [11] of relatively high expression levels in PECs (mean: 89.96 CPM, Figure 5a–c). The mean CPM values in the cell types belonging to CNTs or the CDs (PC and IC) were 1.11, 2.56, and 0.78, respectively (Figure 5c). Notably, in contrast to our previous findings, a considerable level of *GPR126* expression was detected in PT cells (mean: 16.00 CPM). The analysis of the data obtained from patients with AKI and CKD revealed a significant upregulation of *GPR126* expression in both conditions (Figure 5d). In AKI, *GPR126* expression was increased in cells of the TAL and CNT, as well as in PCs and ICs of the CD (Figure 5c). Therefore, the tubular system reacted in a similar way upon AKI as we previously observed in the murine UUO model (Figure 1 and Figure 2). However, the robust increase in *GPR126* expression in PECs was not observed upon AKI as well as CKD. This is due to a very high variability in *GPR126* expression that is already in the kidney samples of healthy living donors (Figure 5c). In contrast to the rat hDTR model (increase in *Gpr126* expression only in PECs, Figure 3), a significant increase in *GPR126* expression could be observed in the CKD kidney samples in the clusters of TAL cells, DCT cells, CNT cells, as well as PCs and ICs (Figure 5c). A comparison of the clusters from the living donor, AKI, and CKD cohorts further indicates that *GPR126* expression is affected in ECs in diseased kidneys (mean LD: 3.79 CPM; mean AKI: 48.54 CPM; mean CKD: 28.42 CPM, Figure 5e). Furthermore, a significant increase in *GPR126* expression could be found in interstitial cells in CKD (Figure 5e).

Collectively, these data indicate that *Gpr126* expression is not only altered in animal models of kidney disease, but also in human kidney disease.

### 3.4. GPR126 Expression Is Increased in PECs inside Glomerular Lesions of Patients with FSGS 

Considering the high variability in the scRNA-seq data, even without *GPR126* expression in any cell type in some patients (Appendix A), we decided to analyze *GPR126* expression in patients based on RNAscope^®^ in situ hybridization. As the rodent model of podocyte depletion and human FSGS share a lot of common pathophysiological features, including podocyte loss followed by the induction of repair mechanisms and, subsequently, glomerular scarring [19,39], and the trend of increased *Gpr126* expression in the glomeruli of patients with FSGS based on the NephroSeq data (Appendix A), this condition was selected to be studied further. We performed RNAscope^®^ in situ hybridization on kidney biopsies of five patients with FSGS, imaging every glomerulus contained within the biopsy (numbers of glomeruli: 25, 25, 6, 14, and 17). One biopsy was excluded due to its low glomerular count (six glomeruli). 

PECs were co-stained for Claudin-1, and the RNAscope^®^ signal intensity within Claudin-1-positive cells was quantified using the same approach used for the animal experiments described earlier (*Figure 2a and Figure 3d*). All glomeruli that showed histologic signs of FSGS (segmental solidification of the glomerular tuft) [36,40] in the adjacent PAS staining were identified and further examined (the numbers of glomeruli with signs of FSGS: 5, 4, 5, and 4). The parietal epithelium was divided into two areas: one area with single-layered PECs, showing no morphological changes from a healthy state, and another area containing multi-layered PECs making up the FSGS lesion. These areas are regarded as the “outside lesion” and “FSGS lesion”, respectively. Interestingly, PECs within the FSGS lesion appeared to show a higher density of RNAscope^®^ signal dots and therefore had a higher expression of *GPR126* mRNA (Figure 6a). This observation was confirmed by the quantification of the percentage of PECs containing three or more signal dots both inside and outside of the FSGS lesion (mean outside lesion: 43.5 ± 7.33; mean FSGS lesion: 65.6 ± 6.20; *p* < 0.01; n = 4) (Figure 6b). It seems that *GPR126* mRNA is upregulated specifically in PECs belonging to FSGS lesions. Interestingly, we could also observe elevated levels of *Gpr126* expression in structures outside the glomerulus. A portion of E-cadherin-negative tubules in the renal cortex—regarded as PTs—showed a marked increase in RNAscope^®^ signal dots (Figure 6c and Appendix A). Tubular damage is a common histologic finding in FSGS and results from the degeneration of the corresponding glomerulus [41], which could explain the restricted affection of tubules observed in our investigation.

Collectively, the data obtained from the biopsies of patients with FSGS show that *GPR126* expression is also increased upon disease in humans, yet the changes are localized to a subpopulation of PECs, namely PECs belonging to glomerular FSGS lesions.

### 3.5. Gpr126 Expression Is Upregulated within Hours upon I/R

Our data show that *Gpr126* expression is markedly increased in PECs in a model of glomerular damage at 42 dpi but not at 21 dpi. To assess whether *Gpr126* expression is generally upregulated late in kidney disease, we analyzed *Gpr126* expression upon I/R over time utilizing Nanostring technology. *Gpr126* expression was not altered 10 min post I/R but significantly upregulated at 6 h after I/R injury in rat kidneys (Figure 7a,b). Expression also appeared elevated at 24 h and at 3 and 5 days, but this difference did not reach the significance level. Eight weeks after I/R, the *Gpr126* expression levels were back to the control levels. To determine whether *Gpr126* expression correlated with kidney function, we correlated the *Gpr126* expression levels with the serum urea levels at 6 h post I/R. This analysis revealed a positive correlation between *Gpr126* expression and serum urea levels (Figure 7a,c). Notably, this correlation persisted at the same level when the data from all time points were included in the analysis (Figure 7d).

Collectively, our data show that *Gpr126* expression is altered in kidney disease. Notably, the location and timing of changes in expression levels as well as the affected cell type depend on the type of injury and the location of the injury. In addition, *Gpr126* expression correlated with disease severity at least in the I/R model of acute kidney injury.

## 4. Discussion

Our work demonstrates that changes in *Gpr126* expression are associated with kidney disease. Increased *Gpr126* expression can be detected in models of tubular and glomerular kidney disease. Notably, cell types in which *Gpr126* expression was upregulated were dependent on the compartment/renal structure that was primarily damaged: in the tubular system as well as in PECs in a mouse model of kidney fibrosis (UUO) and in PECs in a rat model of podocyte depletion (hDTR). Moreover, changes in the *Gpr126* expression levels depended on the type of kidney injury and correlated with disease severity at least in the I/R model of acute kidney injury based on the serum urea levels. Similar alterations in *Gpr126* expression were also found in patients with AKI and CKD, with the distal nephron (CNT and CD) showing the most significant upregulation. Furthermore, *GPR126* is upregulated PECs within glomerular lesions of patients suffering from FSGS. While we characterized the expression of *Gpr126* in different kidney diseases/disease models in detail, it remains unclear whether changes in *Gpr126* expression are causal or drivers of kidney disease by inducing altered cell behavior and actively promoting disease in an attempt to protect the kidney, which might be inefficient due to missing receptor activation, or maybe even only an epiphenomenon. The early upregulation of *Gpr126* expression in the I/R model suggests a role in acute injury. In contrast, the relatively late upregulation in the hDTR model suggests a role in chronic changes or in regeneration. Here, *Gpr126* was expressed in PECs, which, in this model, proliferate after podocyte injury and form glomerular crescents, an observation we also made in human biopsies in FSGS. Notably, aGPCRs have been shown to control cellular processes underlying kidney disease [42,43,44,45,46,47] such as cell polarity [48], mitotic spindle orientation [49], cell migration [48,50], cell aggregation [51], and the transduction of mechanical stimuli [12,52]. Considering the importance of GPCRs as drug targets [6] and the recent finding that the pharmacologic targeting of Adgrg3-mediated Sema3A signaling may provide an approach for AKI treatment [9], we will utilize inducible, conditional Gpr126 knockout mice in the future to determine whether the lack of Gpr126 improves or worsens the disease outcome in different kidney injury models and assess changes in the behavior of *Gpr126*-expressing cells and their neighboring cells. Notably, it has been shown that Gpr126 signaling can be activated in vivo by the peptide FT(23-50) [17] and inhibited by miR27a/b [18]. 

The fact that increased *Gpr126* expression has been observed in distinct experimental models could also suggest that Gpr126 might regulate a general process or cell function in kidney pathology, such as cell proliferation or migration. Notably, renal fibrosis represents the common pathway of most kidney pathologies in the progression of CKD [28], and the level of fibrosis is different in varying injury models, similar to the *Gpr126* expression levels. The pathophysiological changes involved in the progression of renal fibrosis include tubular epithelial cell apoptosis by either mechanical or oxidative stress, the activation of the renin–angiotensin–aldosterone system, NF-κB-mediated inflammation, and fibroblast activation [26,28]. The process of renal fibrosis is regarded as irreversible, and therefore, intensive research is needed to stop or better prevent this process [26,53]. As a model of tubular injury leading to renal fibrosis, the UUO model is widely used to study possible treatments to ameliorate or even reverse this common pathway of renal disease [28]. The rise in hydrostatic pressure leads to an initial dilation of the renal pelvis and the CDs, followed by the distal and proximal nephron segments [28]. In our study, we found *Gpr126* to be highly expressed in urothelium and CD after UUO injury, whereas in the more proximal nephron segments, only the cortical DTs showed a significant increase in *Gpr126* expression. It is possible that this is due to the low expression of *Gpr126* in PTs and DTs under healthy conditions. However, as there is evidence for an involvement of Gpr126 in mechano-dependent signaling [12] and mechanical stress seems to play a major role in UUO-injury [54], the increase in *Gpr126* expression along the injured nephron could also result from different levels of mechanical stress acting on the cells along the nephron, leading to the segment-specific upregulation of *Gpr126*. As patients who suffered tubular damage in an episode of AKI are more likely to develop CKD in the future, it is of key interest to find treatments to ameliorate AKI [55]. Thus, it will be important in the future to determine whether the inhibition or activation of Gpr126 signaling has an effect on AKI-induced fibrosis.

The so-called extension phase in AKI happens after the initial ischemic injury and is characterized by the activation of the inflammatory cascade, leading to further apoptosis and necrosis of renal tubular epithelial cells. Therefore, the interruption of this inflammatory cascade is suggested to be a promising target for the possible treatment of AKI [56]. Notably, Gpr97—a member of the aGPCR family—was shown to exacerbate AKI in a mouse model by mediating the proinflammatory molecule Sema3A [9,57], and there is evidence of an involvement of Gpr126 in inflammatory processes in other diseases, such as a steady-state stimulation of inflammation in gut-associated lymphoid tissue [58]. Furthermore, thrombin- and lipopolysaccharide-mediated inflammation induces the expression of *GPR126* in human umbilical vein endothelial cells in vitro via mitogen-activated protein kinases [59]. Besides having a possible role in inflammation, tissue repair mechanisms triggered by Gpr126 signaling are also imaginable. For example, Gpr126 is critical for the repair process of peripheral nerve damage by ensuring Schwann cell function [13,60]. While a lot of effort has been made to study the pathophysiology of the PT, the glomerulus, and the vasculature in AKI [61], only little is known about the role of the CD in this condition. However, there is emerging evidence that the CD also significantly contributes to the progression of AKI. For example, CD intercalated cells promote renal inflammation through the P2Y_14_ purinergic receptor-mediated production of chemoattractant cytokines [62]. Therefore, it is important to further investigate the pathophysiological role of Gpr126 in the CD, utilizing markers for different cellular processes such as inflammation, proliferation, oriented cell division, and migration.

In addition to our findings in cells belonging to the nephron itself, we found *GPR126* to be upregulated in the ECs of kidneys of patients with AKI and CKD (Figure 5e). This finding fits previous studies reporting that *Gpr126* is expressed in human ECs [2] and in murine ECs of renal blood vessels [12]. As CKD is associated with renal vascular malfunction, which may induce AKI through the affection of the renal tubular blood supply and changes in the glomerular architecture, it will be interesting to assess whether Gpr126 plays a role in this vascular malfunction and maladaptive processes in kidney disease [63].

A common ground across both of our animal models (UUO and hDTR), the scRNA-seq analysis of patients with AKI and CKD, and the analysis of patients with FSGS was the more or less distinct upregulation of *Gpr126* in PECs. Whilst serving as the epithelial layer lining the Bowman´s capsule, PECs are not attributed a particular physiological (transport) function in healthy adult kidneys [64]. In glomerular disease, however, PECs have emerged as a key player, and several mechanisms of PECs interfering with glomerular disease progression are discussed [65]. Upon podocyte damage in FSGS, PECs are activated and express CD44 de novo [66]. Activated PECs are exclusively found in the glomerular sclerosing segments [67], which is curious, as it is these lesions in which we could observe increased *GPR126* expression in our studies (Figure 5). The activated PECs then begin to proliferate, migrate onto the glomerular tuft, and produce an extracellular matrix, crucially shaping the process of glomerulosclerosis [39]. Our data show a marked increase in *Gpr126* expression in PECs in both animal models (UUO and hDTR), which was not apparent in the scRNA-seq analysis of the renal biopsies of patients with AKI and CKD. Regarding the scRNA-seq experiments, it must be mentioned that the data underlie a significant variability. One reason for this might be the different quantity and distribution of cell types contained in each biopsy as a result of the inhomogeneous anatomical arrangement of cell types within the different kidney segments and the variation in the biopsy needle position. This can lead to an underrepresentation of certain cell types in the sample of a certain patient biopsy and can be visualized by the small size or absence of the corresponding cell cluster in the UMAP of the sample (Appendix A). Therefore, the analysis might be limited—especially when analyzing relatively small cell clusters such as PECs.

Considering the limitations of the scRNA-seq data, we analyzed tissue samples of patients with FSGS and also observed an upregulation of *Gpr126* expression in PECs in humans, which was limited to the PECs inside glomerular lesions. FSGS is a leading cause of end-stage renal disease (ESRD) and is among the primary nephrotic diseases that are the most likely to progress to ESRD [36]. In general, FSGS is induced by a malfunction or loss of podocytes, a cell type that has an integral role in maintaining the glomerular filtration barrier [39]. Due to the limited proliferative capacity of podocytes, its damage results in a decrease in the podocyte number, causing proteinuria and glomerulosclerosis [68]. Glomerular damage is regularly followed by an activation of PECs, leading to the proliferation, migration, and extracellular matrix synthesis of these cells [39]. Whether PECs are also able to transdifferentiate into podocytes upon podocyte loss is a topic of intense discussion [39]. Thus, it will be important to determine whether the modulation of Gpr126 signaling has an effect on PEC-mediated lesion formation or the transdifferentiation of PECs into podocytes.

It should also be noted that the association of altered *Gpr126* expression might provide an opportunity for utilizing *Gpr126* expression as a diagnostic marker. In clinical practice, most cases of kidney disease, regardless of the actual cause, can be assigned to either AKI or CKD [69]. The definition and staging of these conditions are proposed and regularly updated by independent guideline development workgroups and mainly rely on functional criteria (AKI: https://kdigo.org/guidelines/acute-kidney-injury/ (accessed on 6 August 2023); CKD: https://kdigo.org/guidelines/ckd-evaluation-and-management/ (accessed on 6 August 2023)). CKD is defined by a decreased GFR for >3 months in combination with an objective measure of kidney damage (primarily albuminuria). The definition of AKI is solely based on changes in kidney function (serum creatinine or urine output) within a short period of time [69]. Following an intense investigation, urinary biomarkers were identified for early detection and diagnosis (e.g., kidney injury molecule 1, neutrophil gelatinase-associated lipocalin, and interleukin 18). However, none of these markers could be implemented into the common diagnostic workflow of AKI [69,70], making it even more important to search for other possible candidates.

## 5. Conclusions

The results shown in this work demonstrate that *Gpr126* expression is altered in kidney disease, namely in AKI and CKD. *Gpr126* expression is affected after tubular as well as glomerular damage. Thus, our data identify the receptor Gpr126 as a potential therapeutic target in kidney disease. In the future, it will be important to determine whether the inhibition or activation of Gpr126 signaling can delay or prevent kidney disease progression or even reverse kidney damage, restoring renal function.

## Figures and Tables

**Figure 1 cells-13-00874-f001:**
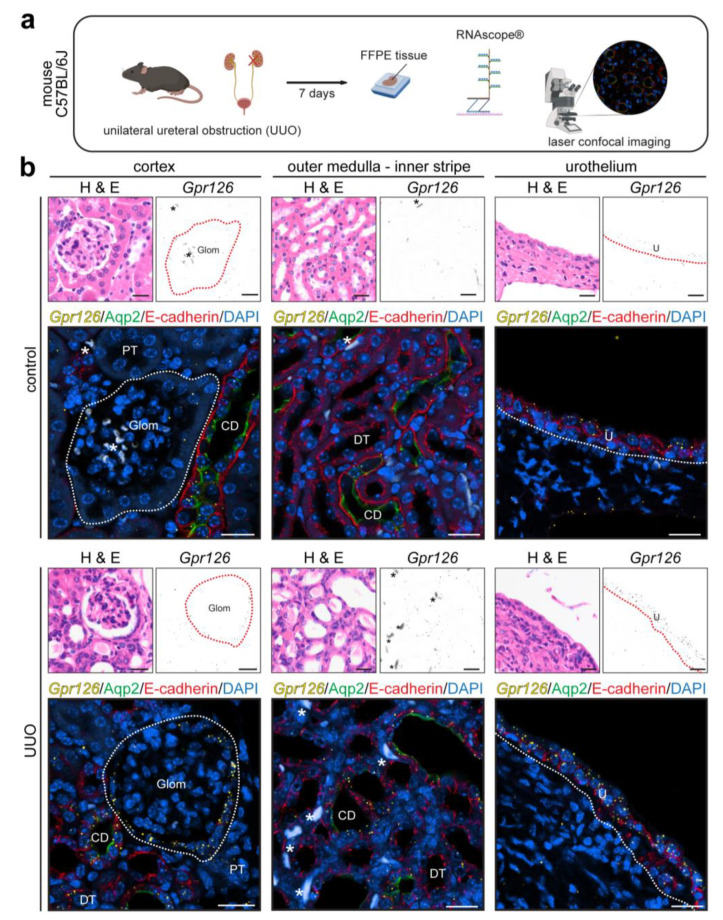
*Gpr126* expression changes in murine model of unilateral ureteral obstruction (UUO). (**a**) UUO was induced by clamping of the left ureter in C57BL/6J mice. After 7 days, FFPE kidney sections were analyzed for *Gpr126* mRNA by RNAscope^®^ in situ hybridization (created with BioRender.com). (**b**) Hematoxylin and eosin (H&E) and RNAscope^®^ staining of both the affected UUO and the contralateral kidney (control). Cortical *Gpr126* expression is increased the strongest in parietal epithelial cells (PECs) lining the glomerular Bowman´s capsule (dotted circle) and collecting duct (CD: E-cadherin-positive/Aqp2-positive). In addition, signal was sometimes also detected in the distal tubule (DT: E-cadherin-positive/Aqp2-negative) and proximal tubule (PT: E-cadherin-negative/Aqp2-negative). In the inner stripe of the outer medulla changes in *Gpr126* expression are most evident in CD. The pelvic urothelium (U: E-cadherin-positive), which, under healthy conditions, expresses more *Gpr126* than any other kidney cell type, also shows a strong increase in *Gpr126* expression. Autofluorescent red blood cells are marked by asterisks (*). Scale bar: 20 µm.

**Figure 2 cells-13-00874-f002:**
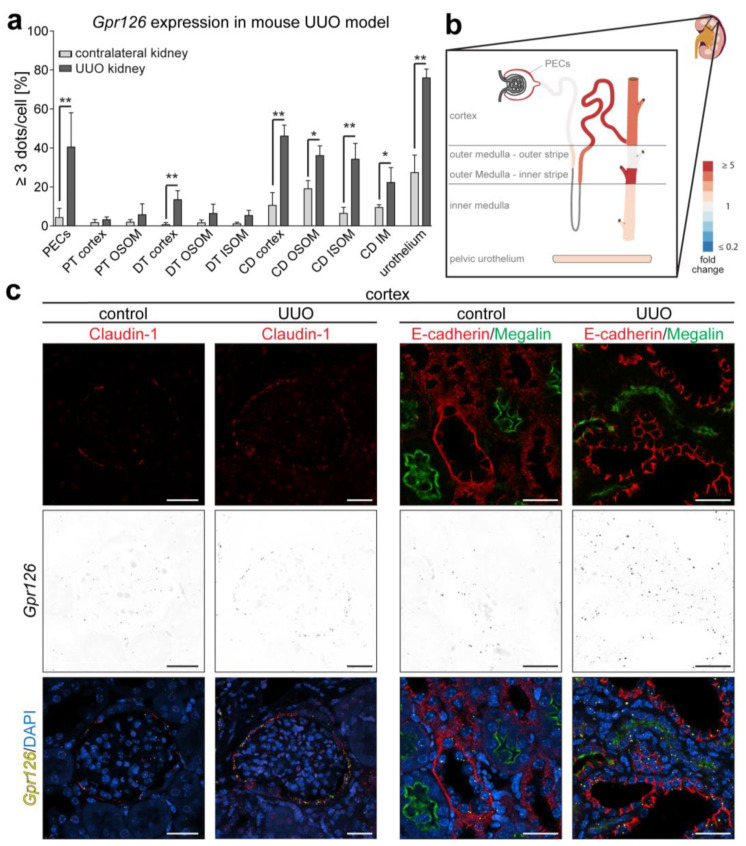
Quantification of *Gpr126* expression changes over different renal cell types and nephron segments. (**a**) Bars show the percentage of cells containing three or more RNAscope^®^ signal dots or at least one dot cluster, comparing the UUO-affected kidney with its contralateral counterpart. Data are shown as means ± SD; n ≥ 3 animals. Statistical analysis: *t*-test. *: *p* < 0.05; **: *p* < 0.01. (**b**) Visualization of *Gpr126* upregulation in mouse UUO model based on quantification data in (**a**). Areas with fold change ≥ 2 in comparison to the contralateral kidney (PECs, PT (OSOM), DT (ISOM, OSOM, and cortex), CD (cortex, ISOM, and IM), and urothelium) are shown in increasingly dark red color (created with smart.servier.com). (**c**) Verification of cell types undergoing changes in *Gpr126* expression. RNAscope^®^ in situ hybridization in the mouse UUO model at day 7 utilizing anti-Claudin-1 (PECs), anti-E-cadherin (distal nephron segments), and anti-Megalin (PT) antibodies. Scale bar: 25 µm. OSOM: outer stripe of outer medulla; ISOM: inner stripe of outer medulla; IM: inner medulla.

**Figure 3 cells-13-00874-f003:**
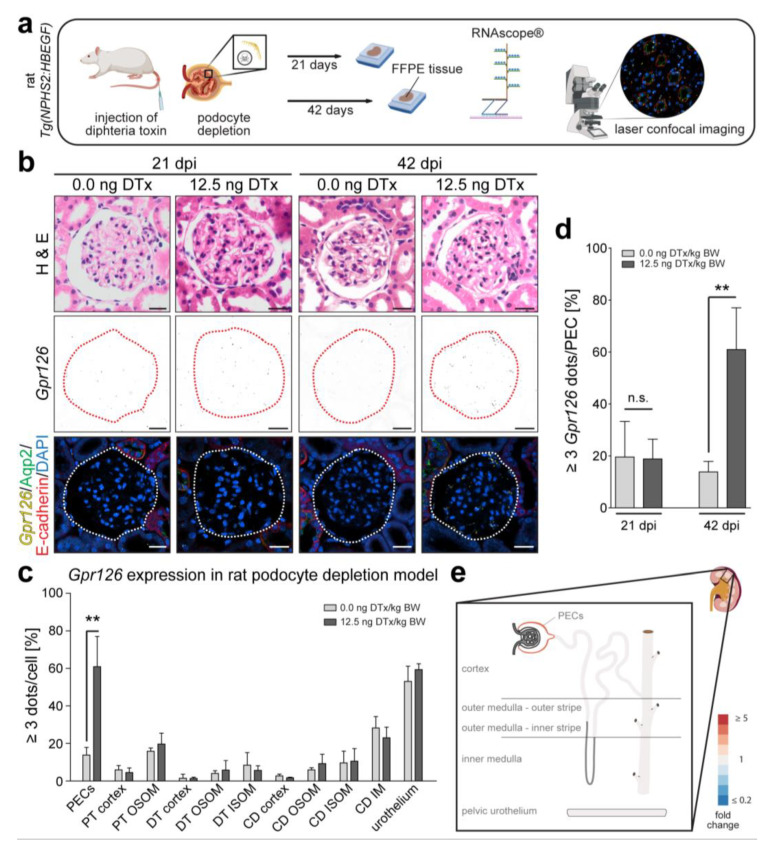
*Gpr126* is upregulated in PECs of rats undergoing podocyte depletion. (**a**) Rats of transgenic line *Tg(NPHS2-HBEGF)*, which expresses human diphteria toxin receptor, specifically in podocytes, received intravenous injections of either 12.5 ng diphteria toxin (DTx)/kg body weight (BW) to induce podocyte depletion or 0 ng DTx/kg BW as control. FFPE kidney tissue was produced after 21 and 42 days post injection (dpi) and analyzed using RNAscope^®^ in situ hybridization (created with BioRender.com). (**b**) H&E and RNAscope^®^ staining of consecutive sections of healthy (0 ng DTx) and diseased (12.5 ng DTx) animals at 21 and 42 dpi. *Gpr126* expression in PECs lining the Bowman’s capsule (dotted line) is unchanged at 21 dpi and increased at 42 dpi. Scale bar: 20 µm. (**c**) Quantification of *Gpr126* expression changes at 42 dpi over different renal cell types and nephron segments (PT; DT; CD; OSOM; ISOM; IM). Bars show percentage of cells containing three or more RNAscope^®^ signal dots or at least one dot cluster, respectively. Apart from upregulation in PECs, no change in *Gpr126* expression can be seen. Data are shown as means ± SD of at least three analyzed animals (n = 3). Statistical analysis was performed using *t*-test. **: *p* < 0.01. (**d**) Quantification of *Gpr126* expression in PECs. At 42 dpi, there is 3.3-fold increase in cells containing three or more RNAscope^®^ signal dots or at least one dot cluster, respectively. At 21 dpi, no differences were observed. Data are shown as means ± SD; n = 4. Statistical analysis: *t*-test. **: *p* < 0.01; n.s.: not significant. (**e**) Visualization of *Gpr126* expression changes upon rat podocyte depletion based on the quantitative data in (**d**). Areas with a fold change ≥ 2 (PECs)—normalized to the expression levels in the contralateral kidney–are shown in increasingly dark red color (created with smart.servier.com).

**Figure 4 cells-13-00874-f004:**
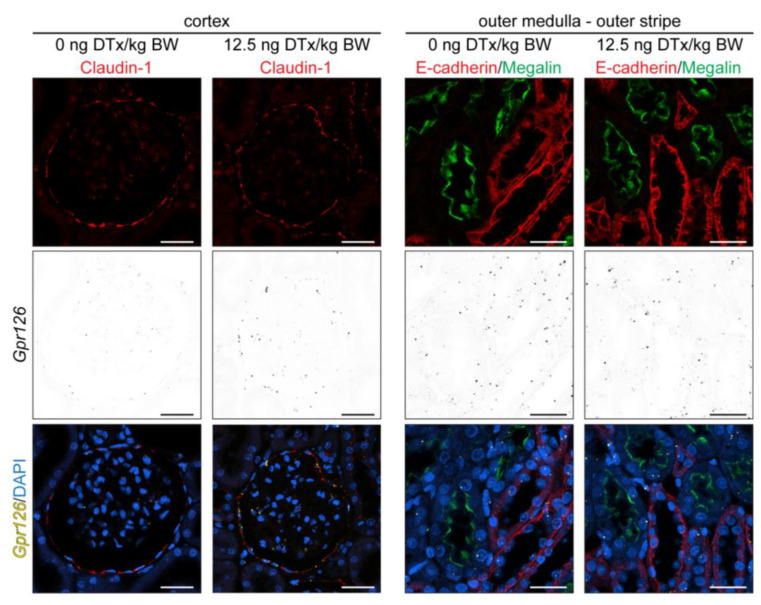
Verification of cell types undergoing changes in *Gpr126* expression in rat podocyte depletion model. RNAscope^®^ in situ hybridization was combined with anti-Claudin-1 antibody staining. The data show that *Gpr126* is upregulated in PECs of *Tg(NPHS2-HBEGF)* rats 42 days after intravenous injection of 12.5 ng diphteria toxin (DTx)/kg body weight (BW). In the outer stripe of the outer medulla, anti-Megalin and anti-E-cadherin staining are mutually exclusive, allowing the identification of the PT, based on the absence of E-cadherin staining. *Gpr126* expression levels, which are higher in distal nephron segments (E-cadherin-positive) than in the PT (Megalin-positive), do not increase in the DTx-treated animals. Scale bar: 25 µm.

**Figure 5 cells-13-00874-f005:**
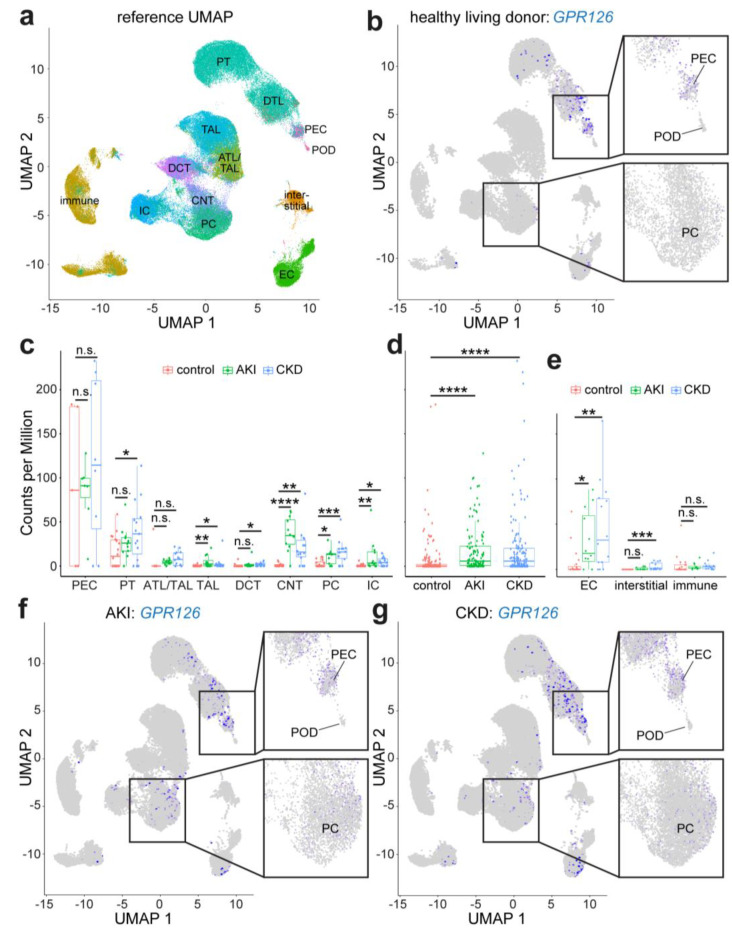
Single-cell RNA-seq analysis of *GPR126* expression in human kidney disease utilizing data from KPMP Kidney Tissue Atlas. (**a**) Reference UMAP of cell clusters (source: https://atlas.kpmp.org/explorer/dataviz, accessed on 18 July 2023). (**b**) UMAP plot of *GPR126* expression for 20 healthy living donors (control). Normalized and scaled UMI (Unique Molecular Identifier) counts are colored in scale ranging from 0 UMI counts (gray) to the highest UMI count (blue). (**c**) Statistical analysis of *GPR126* expression in PECs, PT cells, cells belonging to the ascending thin limb/thick ascending limb (ATL/TAL), thick ascending limb (TAL), or distal convoluted tubule (DCT), connecting tubular cells (CNT), principal cells (PC), and intercalated cells (IC) of the CD from healthy living donors as well as patients with acute kidney injury (AKI) and chronic kidney disease (CKD). (**d**) Statistical analysis of *GPR126* expression in living donor, AKI, and CKD sample. (**e**) Statistical analysis of *GPR126* expression in endothelial cells (EC), interstitial cells, and immune cells. (**f**,**g**) UMAP plot of *GPR126* expression for 12 patients with AKI (**f**) and 17 patients with CKD (**g**). Statistical analysis was performed using Dunn’s test. n.s.: not significant, *: *p* < 0.05, **: *p* < 0.01, ***: *p* < 0.001, and ****: *p* < 0.0001. POD: podocytes.

**Figure 6 cells-13-00874-f006:**
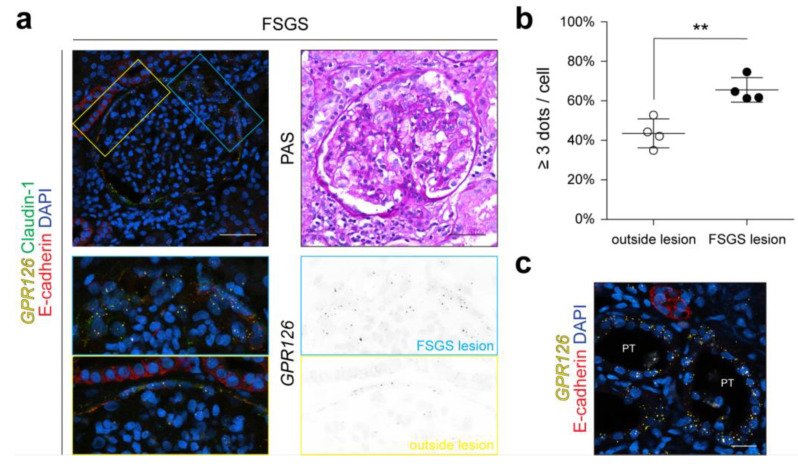
*GPR126* expression is increased in PECs inside glomerular lesions of patients with focal segmental glomerulosclerosis (FSGS). (**a**) RNAscope^®^ in situ hybridization and PAS staining images of human glomerulus affected by primary FSGS. PECs, co-stained with anti-Claudin-1 antibody, show multi-layered appearance inside the crescent, accompanied by a higher expression of *GPR126* compared to the single layered PECs in the remaining glomerulus. Scale bars: 50 µm. (**b**) Quantification of increased expression inside cFSGS lesions compared to the remaining PECs of glomeruli affected by FSGS. All glomeruli with FSGS lesions visible in the biopsy were analyzed. Each dot resembles the mean value ± SD for PECs inside and outside the FSGS lesion in a single biopsy. n = 4 biopsies (from 4 different patients) were analyzed. Statistical analysis was performed using *t*-test. **: *p* < 0.01. (**c**) PPT (E-cadherin-negative) affected by tubular damage in FSGS biopsies show increased *GPR126* expression. Scale bar: 20 µm.

**Figure 7 cells-13-00874-f007:**
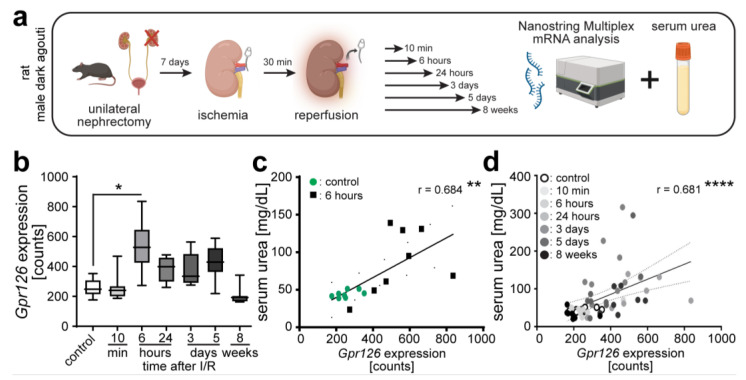
*Gpr126* expression is upregulated within hours upon ischemia/reperfusion injury (I/R). (**a**) Male dark agouti rats were first unilaterally nephrectomized. This kidney was used as control. To induce I/R, the renal artery was disconnected from the blood supply for 30 min with an atraumatic clamp. Animals were sacrificed 10 min, 6 h, 24 h, 3 days, 5 days, and 8 weeks after reperfusion, and serum was collected to determine serum urea as surrogate marker of kidney function (created with BioRender.com). (**b**) *Gpr126* expression measured using Nanostring Multiplex mRNA analysis. ANOVA and Dunn test. (**c**) Correlation of serum urea with *Gpr126* expression at 6 h post I/R or (**d**) all investigated time points. Spearman test. *: *p* < 0.05; **: *p* < 0.01; and **** *p* < 0.0001.

## Data Availability

The data are all included in the article and Appendix A. No omics data were generated.

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
