# Peer review of "Adhesion G Protein-Coupled Receptor Gpr126 (Adgrg6) Expression Profiling in Diseased Mouse, Rat, and Human Kidneys"

_cells, 2024, doi:10.3390/cells13100874_

Round 1
Reviewer 1 Report
Comments and Suggestions for Authors
The manuscript "Adhesion GPCR Gpr126 (Adgrg6) Expression Profiling in Diseased Mouse, Rat, and Human Kidney" presents an analysis of the expression of Gpr126 in experimental models of kidney damage and samples from patients. The work includes a detailed histopathological analysis of the cellular expression patterns of Gpr126. The data presented are consistent with the conclusions that a) Gpr126 expression is altered in models of kidney disease and patients with AKI and CKD, and b) Gpr126 expression is affected after tubular and glomerular damage. I think the manuscript provides new information that can interest nephrologists and researchers in this area. Nevertheless, some aspects should be considered.
Major points
1.- The manuscript is mainly descriptive. It does not provide information on the relationship of Gpr126 expression with the magnitude of the kidney damage or its functional impact. Similarly, there is no information on the time courses, i.e., it does not address if Gpr126 expression precedes or accompanies the histological and functional damages. There is no information on the magnitude of expression concerning the degree of kidney dysfunction.
2. The changes were apparent in some of the models employed, whereas in others they were marginal. These limitations, and those mentioned in point 1, should be clearly stated and discussed.
3.- In several manuscript sections, the authors repeatedly indicate that Gpr126 is a possible therapeutic target in kidney disease. The data indicate an association between Gpr126 expression and kidney damage; it could be suggested as a general marker of kidney disease. However, there is no evidence that Gpr126 might play a role in the pathogenesis or maintenance of kidney damage. The association does not indicate any role in causation. The fact that such increased expression is observed in distinct experimental models argues against any specificity, suggesting that it is a general process in all of them. It could be an epiphenomenon associated with hemodynamic changes or mechanical stress, as indicated by the authors, rather than a causative factor.
4.- It could be interesting to know if such increased Gpr126 expression is observed in diseases that include some degree of kidney damage (hypertension, type 2 diabetes mellitus, urinary tract infections, among others).
Reviewer 2 Report
Comments and Suggestions for Authors
Authors investigated Gpr126 expression in diseased kidneys of mice, rats, and humans. Authors found that the expression level of Gpr126 is associated with kidney diseases. The manuscript presents interesting and critical observation on the potentially significant function of Grp126 in kidney diseases. However, no evidence verifies that the expressed Gpr126 participates in kidney diseases in any way. That is, it remains unclear whether Gpr126 expression is the cause or the consequence of kidney diseases. Hence, the manuscript raised several questions before the consideration of the publications:
Main points:
· Results in the manuscript shows a correlation between the expression level of Gpr126 and kidney diseases, instead of a “cause-to-result” demonstration. This needs to be rephrased properly throughout of the manuscript.
· Is it feasible to examine the activation of Gpr126 signaling pathway, such as some key downstream signaling molecules of Grp126, in any organism of mice, rat, or human?
· Quantitation of Figure 4.
· Manuscript requires a thorough copy-editing.
Minor points:
· A brief explanation of “A direct link between aGPCRs and kidney development/disease has been shown for Celsr1 (ureteric bud (UB) branching) [8] and Adgrg3 (AKI) [9]” would be helpful for readers to understand “pharmacologic targeting of Adgrg3-mediated Sema3A signaling.”
· A brief explanation of expansion phase of AKI, vascular maladaptation, and any other special terms used without a brief explanation of the relevance.
· First time use of a special term or abbreviation requires a full name and favorably with a explanation of the relevance.
Comments on the Quality of English LanguageManuscript needs a thorough copy-editing.
Round 2
Reviewer 2 Report
Comments and Suggestions for Authors
Authors have adequately addressed comments.